# Hierarchical tissue organization as a general mechanism to limit the accumulation of somatic mutations

Imre Derényi[1] & Gergely J. Szöllősi[2]

How can tissues generate large numbers of cells, yet keep the divisional load (the number of divisions along cell lineages) low in order to curtail the accumulation of somatic mutations and reduce the risk of cancer? To answer the question we consider a general model of hierarchically organized self-renewing tissues and show that the lifetime divisional load of such a tissue is independent of the details of the cell differentiation processes, and depends only on two structural and two dynamical parameters. Our results demonstrate that a strict analytical relationship exists between two seemingly disparate characteristics of self-renewing tissues: divisional load and tissue organization. Most remarkably, we find that a sufficient number of progressively slower dividing cell types can be almost as efficient in minimizing the divisional load, as non-renewing tissues. We argue that one of the main functions of tissue-specific stem cells and differentiation hierarchies is the prevention of cancer.

[1] Department of Biological Physics, ELTE-MTA 'Lendulet' Biophysics Research Group, Eötvös University, Pázmány P. stny. 1A, Budapest H-1117, Hungary. [2] Department of Biological Physics, ELTE-MTA 'Lendulet' Evolutionary Genomics Research Group, Eötvös University, Pázmány P. stny. 1A, Budapest H-1117, Hungary. Correspondence and requests for materials should be addressed to I.D. (email: derenyi@elte.hu) or to G.J.S. (email: ssolo@elte.hu).

In each multicellular organism a single cell proliferates to produce and maintain tissues comprised of large populations of differentiated cell types. The number of cell divisions in the lineage leading to a given somatic cell governs the pace at which mutations accumulate[1]. The resulting somatic mutational load determines the rate at which unwanted evolutionary processes, such as cancer development, proceed[2–4]. To produce $N$ differentiated cells from a single precursor cell the theoretical minimum number of cell divisions required along the longest lineage is $\log_2(N)$. To achieve this theoretical minimum, cells must divide strictly along a perfect binary tree of height $\log_2(N)$ (Fig. 1a). In multicellular organisms such differentiation typically takes place early in development. It is responsible for producing the cells of non-renewing tissues (for example, primary oocytes in the female germ line[1,5]) and the initial population of stem cells in self-renewing tissues (for example, hematopoietic stem cells[6–8] or the spermatogonia of the male germ line[1,5]).

In self-renewing tissues, which require a continuous supply of cells, divisions along a perfect binary tree are unfeasible. Strictly following a perfect binary tree throughout the lifetime of the organism would require extraordinarily elaborate scheduling of individual cell divisions to ensure tissue homoeostasis[9], and would be singularly prone to errors (for example, the loss of any single cell would lead to the loss of an entire branch of the binary tree). Instead, to compensate for the continuous loss of cells, mechanisms have evolved to replenish the cell pool throughout the organism's lifetime[10]. In most multicellular organisms hierarchically organized tissue structures are utilized. At the root of the hierarchy are a few tissue-specific stem cells defined by two properties: self-replication and the potential for differentiation[11,12]. During cell proliferation cells can differentiate and become increasingly specialized toward performing specific functions within the hierarchy, while at the same time losing their stem cell-like properties (Fig. 1b). A classic example is the hematopoietic system[13,14], but other tissues such as skin[15] or colon[16,17] are also known to be hierarchically organized. Identifying each level of the hierarchy, however, can be difficult, especially if the cells at different levels are only distinguished by their environment, such as their position in the tissue (for

example, the location of the transit-amplifying cells along intestinal crypts). As a result, information on the details of differentiation hierarchies is incomplete[18–20].

Nonetheless, in a recent paper, Tomasetti and Vogelstein[21] gathered available information from the literature and investigated the determinants of cancer risk among tumours of different tissues. Examining cancers of 31 different tissues they found that the lifetime risk of cancers of different types is strongly correlated with the total number of divisions of the normal self-replicating cells. Their conclusion that the majority of cancer risk is attributable to bad luck[21] arguably results from a misinterpretation of the correlation between the logarithms of two quantities[22,23]. However, regardless of the interpretation of the correlation, the data display a striking tendency: the dependence of cancer incidence on the number of stem cell divisions is sub-linear, that is, a 100 fold increase in the number of divisions only results in a 10 fold increase in incidence. This indicates that tissues with a larger number of stem cell divisions (typically larger ones with rapid turnover, for example, the colon) are relatively less prone to develop cancer. This is analogous to the roughly constant cancer incidence across animals with vastly different sizes and life-spans (Peto's paradox), which implies that large animals (for example, elephants) possess mechanisms to mitigate their risk relative to smaller ones (for example, mice)[24–26].

What are the tissue-specific mechanisms that explain the differential propensity to develop cancer? It is clear that stem cells that sustain hierarchies of progressively differentiated cells are well positioned to provide a safe harbour for genomic information. Qualitative arguments suggesting that hierarchically organized tissues may be optimal in reducing the accumulation of somatic mutations go back several decades[27]. As mutations provide the fuel for somatic evolution (including not only the development of cancer, but also tissue degeneration, aging, germ line deterioration and so on) it is becoming widely accepted that tissues have evolved to minimize the accumulation of somatic mutations during the lifetime of an individual[27]. The potential of hierarchical tissues to limit somatic mutational load simply by reducing the number of cell divisions along cell lineages, however, has not been explored in a mathematically rigorous way. Here, we discuss this most fundamental mechanism by which hierarchical tissue organization can curtail the accumulation of somatic mutations. We derive simple and general analytical properties of the divisional load of a tissue, which is defined as the number of divisions its constituent cells have undergone along the longest cell lineages, and is expected to be proportional to the mutational load of the tissue.

Models conceptually similar to ours have a long history[7,28–33], going back to Loeffler and Wichman's work on modelling hematopoietic stem cell proliferation[28], and several qualitative arguments have been made suggesting why hierarchically organized tissues may be optimal in minimizing somatic evolution. In a seminal contribution Nowak et al.[29] showed that tissue architecture can contribute to the protection against the accumulation of somatic mutations. They demonstrated that the rate of somatic evolution will be reduced in any tissue where geometric arrangement or cellular differentiation induce structural asymmetries such that mutations that do not occur in stem cells tend to be washed out of the cell population, slowing down the rate of fixation of mutations. Here, we begin where Nowak et al.[29] left off: aside of structural asymmetry, we consider a second and equally important aspect of differentiation, the dynamical asymmetry of tissues, that is, the uneven distribution of divisional rates across the differentiation hierarchy.

More recently a series of studies have investigated the dynamics of mutations in hierarchical tissues with dynamical

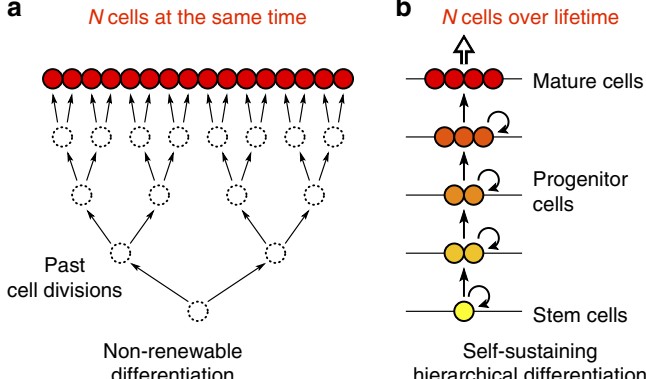

**Figure 1 | Differentiation in non-renewing versus self-renewing tissues.** (**a**) To produce $N$ mature cells from a single precursor with a minimum number of cell divisions, $\log_2(N)$, strict division along a perfect binary tree is necessary. In multicellular organisms such 'non-renewable' differentiation typically takes place early in development. (**b**) However, in self-renewing tissues, where homoeostasis requires a continuous supply of cells, a small population of self-replicating tissue-specific stem cells sustain a hierarchy of progressively differentiated and larger populations of cell types, with cells of each type being continuously present in the tissue.

asymmetry[31–33] and found that hierarchical tissue organization can (i) suppress single[32] as well as multiple mutations[33] that arise in progenitor cells and (ii) slow down the rate of somatic evolution towards cancer[31] if selection on mutations with non-neutral phenotypic effects is also taken into account. The epistatic interactions between individual driver mutations are, however, often unclear and show large variation among cancer types. The fact that the majority of cancers arise without a histologically discernible premalignant phase indicates strong cooperation between driver mutations, suggesting that major histological changes may not take place until the full repertoire of mutations is acquired[34]. For this reason, here we do not consider selection between cells, but rather, focus only on the pace of the accumulation of somatic mutations in tissues, which provide the fuel for somatic evolution.

The uneven distribution of divisional rates considered by Werner et al.[32,33] followed a power law, however, this distribution was taken for granted without prior justification. Their focus was instead on 'reproductive capacity', an attribute of a single cell corresponding to the number of its descendants, which is conceptually unrelated to our newly introduced 'divisional load', which characterizes the number of cell divisions along the longest cell lineages of the tissue. Here we show mathematically, to the best of our knowledge for the first time, that the minimization of the divisional load in hierarchical differentiation indeed leads to power law distributed differentiation rates.

More generally, evolutionary thinking is becoming an indispensable tool to understand cancer, and even to propose directions in the search for treatment strategies[35]. Models that integrate information on tissue organization have not only provided novel insight into cancer as an evolutionary process[27,36,37], but have also produced direct predictions for improved treatment[38–40]. The simple and intuitive relations that we derive below have the potential to further this field of research by providing quantitative grounds for the deep connection between organization principles of tissues and disease prevention and treatment.

According to our results, the lifetime divisional load of a hierarchically organized tissue is independent of the details of the cell differentiation processes. We show that in self-renewing tissues hierarchical organization provides a robust and nearly ideal mechanism to limit the divisional load of tissues and, as a result, minimize the accumulation of somatic mutations that fuel somatic evolution and can lead to cancer. We argue that hierarchies are how the tissues of multicellular organisms keep the accumulation of mutations in check, and that populations of cells currently believed to correspond to tissue-specific stem cells may in general constitute a diverse set of slower dividing cell types[6,41]. Most importantly, we find that the theoretical minimum number of cell divisions can be very closely approached: as long as a sufficient number of progressively slower dividing cell types towards the root of the hierarchy are present, optimal self-sustaining differentiation hierarchies can produce $N$ terminally differentiated cells during the course of an organism's lifetime from a single precursor with no more than $\log_2(N) + 2$ cell divisions along any lineage.

## Results

**Divisional load of cell differentiation hierarchies.** To quantify how many times the cells of self-renewing tissues undergo cell divisions during tissue development and maintenance, we consider a minimal generic model of hierarchically organized, self-sustaining tissue. According to the model, cells are organized into $n + 1$ hierarchical levels based on their differentiation state. The bottom level (level 0) corresponds to tissue-specific stem cells, higher levels represent progressively differentiated progenitor cells, and the top level (level $n$) is comprised of terminally differentiated cells (Fig. 2a). The number of cells at level $k$ in fully developed tissue under normal homoeostatic conditions is denoted by $N_k$. During homoeostasis cells at levels $k < n$ can differentiate (that is, produce cells for level $k + 1$) at a rate $\delta_k$, and have the potential for self-replication. At the topmost $k = n$ level of the hierarchy terminally differentiated cells can no longer divide and are expended at the same rate $\delta_{n-1}$ that they are produced from the level below. The differentiation rates $\delta_k$ are defined as the total number of differentiated cells produced by the $N_k$ cells of level $k$ per unit time. The differentiation rate of a single cell is, thus $\delta_k / N_k$.

In principle five microscopic events can occur with a cell: (i) symmetric cell division with differentiation, (ii) asymmetric cell division, (iii) symmetric cell division without differentiation, (iv) single cell differentiation and (v) cell death (Fig. 2b). Our goal is to determine the optimal tissue organization and dynamics that minimize the number of cell divisions that the cells undergo until they become terminally differentiated. For this reason cell death, except for the continuous expenditure of terminally differentiated cells, is disallowed as it can only increase the number of divisions. We note, however, that cell death with a rate proportional to that of cell divisions would simply result in a proportionally increased divisional load and, thus, would have no effect on the optimum.

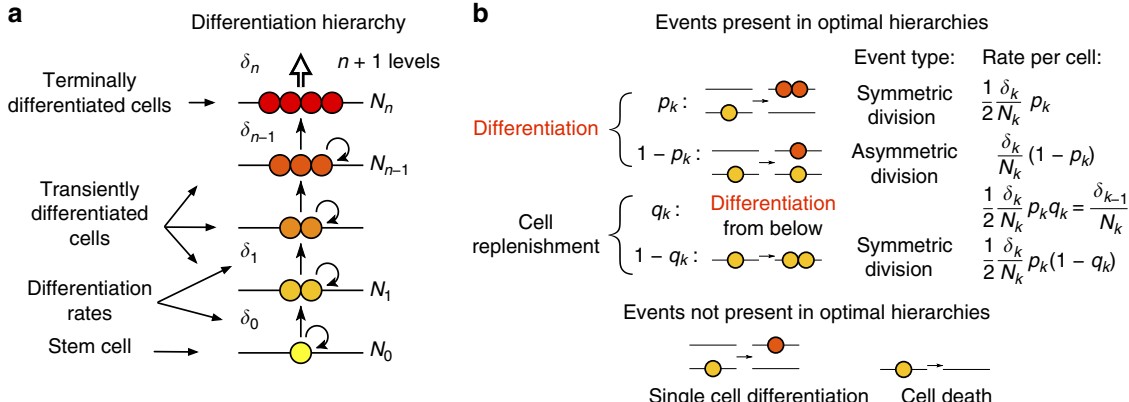

**Figure 2 | Hierarchical cell differentiation in self-renewing tissue.** (**a**) A model tissue produces terminally differentiated cells through $n$ intermediate levels of partially differentiated cells. (**b**) Five microscopic events can occur with a cell: (i) symmetric cell division with differentiation, (ii) asymmetric cell division, (iii) symmetric cell division without differentiation, (iv) single cell differentiation and (v) cell death. To the right of each type of event present in optimal hierarchies we give the corresponding per cell rate that is used to derive equation (2).

Similarly, we also disregard single cell differentiation, because if it is rare enough (that is, its rate is smaller than the asymmetric cell division rate plus twice the rate of symmetric cell division without differentiation) then it can be absorbed in cell divisions with differentiation; otherwise it would merely delegate the replication burden down the hierarchy towards the less differentiated and supposedly less frequently dividing cells, and would be sub-optimal.

Two of the remaining three microscopic events involve differentiation. If we denote the fraction of differentiation events that occur via symmetric cell division at level $k$ by $p_k$, then the rate of symmetric cell division at level $k$ can be written as $p_k \delta_k / 2$ (the division by two accounts for the two daughter cells produced by a single division), while the rate of asymmetric cell division is $(1 - p_k)\delta_k$. Symmetric cell division with differentiation leaves an empty site at level $k$, which will be replenished either (i) by differentiation from the level below or (ii) by division on the same level. Assuming the first case and denoting the fraction of replenishment events that occur by differentiation from the level below by $q_k$, the combined rate of the contributing processes (asymmetric cell division and symmetric cell division with differentiation from the level below) can be written as $q_k p_k \delta_k / 2$. By definition this is equal to $\delta_{k-1}$, the differentiation rate from level $k - 1$, leading to the recursion relation

$$\delta_{k-1} = \delta_k p_k q_k / 2. \qquad (1)$$

Alternatively, if replenishment occurs by cell division on the same level $k$, that is, as a result of symmetric cell division without differentiation, the corresponding rate is $(1 - q_k)p_k \delta_k / 2$.

To keep track of how cell divisions accumulate along cell lineages during tissue renewal, we introduce the divisional load $D_k(t)$ for each level separately defined as the average number of divisions that cells at level $k$ have undergone by time $t$ since the stem cell level was created at time zero.

Using the rates of the microscopic events (also shown in Fig. 2b), considering that each division increases the accumulated number of divisions of both daughter cells by one, and taking into account the divisional loads that the departure of cells take and the arrival of cells bring, the following mean-field differential equation system can be formulated for the time evolution of the total divisional load $(D_k N_k)$ of levels $k < n$ of a fully developed tissue:

$$\dot{D}_k N_k = - \frac{\delta_k}{2} p_k D_k + \delta_k (1 - p_k)$$
$$+ \frac{\delta_k}{2} p_k [q_k (D_{k-1} + 1) + (1 - q_k)(D_k + 2)]. \qquad (2)$$

Because stem cells cannot be replenished from below we have $q_0 = 0$. The terminal level $k = n$ can be included in the system of equations by specifying $p_n = q_n = 1$ and formally defining $\delta_n = 2\delta_{n-1}$.

The above equations are valid when each level $k$ contains the prescribed number of cells $N_k$ of a fully developed, homoeostatic tissue and, therefore, do not directly describe the initial development of the tissue from the original stem cells. This shortcoming can, however, be remedied by introducing virtual cells that at the initial moment ($t = 0$) fill up all $k > 0$ levels. As the virtual cells gradually differentiate to higher levels of the hierarchy, they are replaced by the descendants of the stem cells. Tissue development is completed when the non-virtual descendants of the initial stem cell population fill the terminally differentiated level for the first time, expelling all virtual cells. Using this approach the initial development of the tissue is assumed to follow the same dynamics as the self-renewal of the fully developed tissue. Even though cell divisions in a developing tissue might occur at an elevated pace, such differences in the overall pace of cell divisions (along with any temporal variation in the tissue dynamics) are irrelevant, as long as only the relation between the number of cell divisions and the number of cells generated are concerned.

Using the recursion relation the above differential equation system simplifies to

$$\dot{D}_k N_k = (\delta_k - \delta_{k-1}) - \delta_{k-1}(D_k - D_{k-1}), \qquad (3)$$

revealing that the average number of cell divisions is independent of both the fraction of symmetric division $p_k$ in differentiation, and the fraction of differentiation $q_k$ in replenishment.

From any initial condition $D_k(t)$ converges to the asymptotic solution

$$D_k(t) = t \frac{\delta_0}{N_0} + D_k^0, \qquad (4)$$

which shows that the divisional load of the entire tissue grows linearly according to the differentiation rate of the stem cells ($t\delta_0 / N_0$), and the progenitor cells at higher levels of the hierarchy have an additional load $\left(D_k^0\right)$ representing the number of divisions having led to their differentiation. By definition, the additional load of the stem cells $\left(D_0^0\right)$ is zero. The convergence involves a sum of exponentially decaying terms, among which the slowest one is characterized by the time scale

$$\tau_k^{\mathrm{tr}} = \sum_{l=1}^{k} \frac{N_l}{\delta_{l-1}}, \qquad (5)$$

which can be interpreted as the transient time needed for the cells at level $k$ to reach their asymptotic behaviour. $\tau_k^{\mathrm{tr}}$ can also be considered as the transient time required for the initial development of the tissue up to level $k$. The rationale behind this is that during development the levels of the hierarchy become populated by the descendants of the stem cells roughly sequentially, and the initial population of level $l$ takes about $N_l / \delta_{l-1}$ time after level $l - 1$ has become almost fully populated.

Plugging the asymptotic form of $D_k(t)$ into the system of differential equations and prescribing $D_0^0 = 0$, the constants $D_k^0$ can be determined, and expressed as

$$\begin{aligned} D_k^0 &= \sum_{l=1}^{k} \frac{\delta_l - \delta_{l-1}}{\delta_{l-1}} - \delta_0 \sum_{l=1}^{k} \frac{N_l}{\delta_{l-1}} \\ &= \sum_{l=1}^{k} (\gamma_l - 1) - \frac{\delta_0}{N_0} \tau_k^{\mathrm{tr}}, \end{aligned} \qquad (6)$$

where we have introduced the ratios

$$\gamma_k = \frac{\delta_k}{\delta_{k-1}} = \frac{2}{p_k q_k} \geq 2 \qquad (7)$$

between any two subsequent differentiation rates. The asymptotic solution then becomes

$$D_k(t) = \frac{\delta_0}{N_0} \left( t - \tau_k^{\mathrm{tr}} \right) + \sum_{l=1}^{k} (\gamma_l - 1). \qquad (8)$$

This simple formula, which describes the accumulation of the divisional load along the levels of a hierarchically organized tissue, is one of our main results.

**Differentiation hierarchies that minimize divisional load.** The number of mutations that a tissue allows for its constituent cells to accumulate can be best characterized by the expected number of mutations accumulated along the longest cell lineages. On average, the longest lineage corresponds to the last terminally differentiated cell that is produced by the tissue at the end of the lifetime of the organism. Therefore, as the single most important characteristics of a hierarchically organized tissue, we define its lifetime divisional load, $D$, as the divisional load of its last terminally differentiated cell. If the total number of terminally

differentiated cells produced by the tissue during the natural lifetime of the organism per stem cell is denoted by $N$, then the lifetime of the organism can be expressed as $t_{\text{life}} = \tau^{\text{tr}}_{n-1} + N_0 N / \delta_{n-1}$, where the first term is the development time of the tissue up to level $n-1$, and the second term is the time necessary to generate all the $N_0 N$ terminally differentiated cells by level $n-1$ at a rate of $\delta_{n-1}$. Because the last terminally differentiated cell is the result of a cell division at level $n-1$, its expected divisional load, $D$, is the average divisional load of level $n-1$ increased by 1:

$$D = D_{n-1}(t_{\text{life}}) + 1 = N \frac{\delta_0}{\delta_{n-1}} + \sum_{l=1}^{n-1} (\gamma_l - 1) + 1$$

$$= N \prod_{l=1}^{n-1} \frac{1}{\gamma_l} + \sum_{l=1}^{n-1} (\gamma_l - 1) + 1. \qquad (9)$$

Note that the complicated $\tau^{\text{tr}}_{n-1}$ term drops out of the formula. A remarkable property of $D$ is that it depends only on two structural and two dynamical parameters of the tissue. The two structural parameters are the total number of the terminally differentiated cells produced by the tissue per stem cell, $N$, and the number of the hierarchical levels, $n$. The two dynamical parameters are the product and sum of the ratios of the differentiation rates, $\gamma_k$. The lifetime divisional load neither depends on most of the microscopic parameters of the cellular processes, nor on the number of cells at the differentiation levels.

For fixed $N$ and $n$ the ratios $\gamma_k^*$ of the differentiation rates that minimize the lifetime divisional load $D$ can be easily determined by setting the derivatives of $D$ with respect to the ratios $\gamma_k$ to zero, resulting in

$$\gamma_k^* = N \prod_{l=1}^{n-1} \frac{1}{\gamma_l^*}. \qquad (10)$$

This expression shows that $\gamma_k^*$ is identical for all intermediate levels ($0 < k < n$) and, therefore, can be denoted by $\gamma^*$ without a subscript. This uniform ratio can then be expressed as

$$\gamma^* = N^{1/n}, \qquad (11)$$

as long as the condition $\gamma^* \geq 2$ holds, that is, when $n \leq \log_2(N)$. For $n \geq \log_2(N)$, however, the ratio has to take the value of

$$\gamma^* = 2. \qquad (12)$$

Plugging $\gamma^*$ into equation (9) results in

$$D^* = n\left(N^{1/n} - 1\right) + 2 \qquad (13)$$

for $n \leq \log_2(N)$ and

$$D^* = N\left(\frac{1}{2}\right)^{n-1} + n \qquad (14)$$

for $n \geq \log_2(N)$. Equation (13) is a monotonically decreasing function of $n$, while equation (14) has a minimum at

$$n_{\text{opt}} = \log_2(N) + 1 + \log_2(\ln 2) \approx \log_2(N) + 0.471 \qquad (15)$$

levels. This $n_{\text{opt}}$ together with the ratio

$$\gamma^*_{\text{opt}} = 2 \qquad (16)$$

represent the optimal tissue-structure in the sense that it minimizes the lifetime divisional load of a self-renewing tissue, yielding

$$D^*_{\text{opt}} = \log_2(N) + 1 + \log_2(\ln 2) + 1/\ln 2$$

$$\approx \log_2(N) + 1.914. \qquad (17)$$

Note that under this optimal condition the divisional rate of the stem cell level is very low: in a mature tissue (that is, after the tissue has developed) the expected number of divisions of a stem cell, which is equivalent to the expected number of differentiation to level 1 per stem cell is only $(\delta_0/N_0)(N_0 N / \delta_{n-1}) = 1/\ln 2 \approx 1.44$.

**Implications of the analytical results**. Remarkably, $D^*_{\text{opt}}$ corresponds to less than two cell divisions in addition to the theoretical minimum of $\log_2(N)$, achievable by a series of divisions along a perfect binary tree characteristic of non-renewing tissues. In other words, in terms of minimizing the number of necessary cell divisions along cell lineages, a self-renewing hierarchical tissue can be almost as effective as a non-renewing one. Consequently, hierarchical tissue organization with a sufficient number of hierarchical levels provides a highly adaptable and practically ideal mechanism not only for ensuring self-renewability but also keeping the number of cell divisions near the theoretical absolute minimum.

An important result of our mathematical analysis is that it provides a simple and mathematically rigorous formula (equations (13 and 14), and Fig. 3) for the lower limit of the lifetime divisional load of a tissue for a given number of hierarchical levels and a given number of terminally differentiated cells descending from a single stem cell. This lower limit can be reached only with a power law distribution of the differentiation rates (that is, with a uniform ratio between the differentiation rates of any two successive differentiation levels), justifying the assumptions of the models by Werner et al.[32,33].

In the optimal scenario, where $\gamma_k = \gamma^*_{\text{opt}} = 2$, the recursion relation imposes $p_n = q_n = 1$, thereby, all cell divisions must be symmetric and involve differentiation. This is a shared feature with non-renewable differentiation, which is the underlying reason, why the number of cell divisions of the optimal self-renewing mechanism can closely approach the theoretical minimum.

As a salient example of self-renewing tissues, let us consider the human skin. Clonal patches of skin are of the order of square millimetres in size[42], the top layer of skin, which is renewed daily, is composed of approximately a thousand cells per square

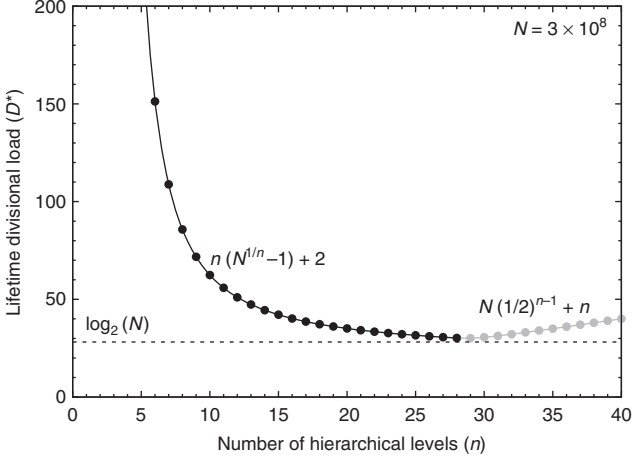

**Figure 3 | The lower limit of the lifetime divisional load as a function of the number of hierarchical levels.** The black and grey solid lines (with filled circles at integer values of $n$) show the lower limit of the lifetime divisional load of a tissue, $D^*$, as a function of the number of hierarchical levels, $n$, for $n \leq \log_2(N)$ and $n \geq \log_2(N)$, respectively. The theoretical minimum, $\log_2(N)$, achievable by a series of divisions along a perfect binary tree characteristic of non-renewing tissues, is displayed with a dashed line. Here we have assumed $N = 3 \times 10^8$ roughly corresponding to the number of cells shed by a few square millimetres of human skin that is sustained by a single stem cell.

millimetre[43]. If we assume that a $10\,mm^2$ patch is maintained by a single stem cell for 80 years, this corresponds to about $N = 3 \times 10^8$ cells. As Fig. 3 demonstrates, the $D^*$ versus $n$ curve becomes very flat for large values of $n$, indicating that in a real tissue the number of hierarchical levels can be reduced by at least a factor of two from the optimal value, without significantly compromising the number of necessary cell divisions along the cell lineages.

It is a question how the total number of terminally differentiated cells ($N_0 N$) produced by the tissue during the natural lifetime of the organism can be best partitioned into the number of tissue-specific stem cells ($N_0$) and the number of terminally differentiated cells per stem cell ($N$). The initial generation of the stem cells along a binary tree requires $\log_2(N_0)$ divisions. The production of the terminally differentiated cells in a near-optimal hierarchy requires about $\log_2(N_0)$ divisions. Their sum, which is about $\log_2(N_0)$, depends only on the total number of terminally differentiated cells, irrespective of the number of stem cells. This means, that the minimization of the divisional load poses no constraint on the number of stem cells. However, since both maintaining a larger number of differentiation levels and keeping the differentiation hierarchy closer to optimum involve more complicated regulation, we suspect that a relatively large stem cell pool is beneficial, especially as a larger stem cell population can also be expected to be more robust against stochastic extinction, population oscillation, and injury.

## Discussion

In general, how closely the hierarchical organization of different tissues in different organisms approaches the optimum described above depends on (i) the strength of natural selection against unwanted somatic evolution, which is expected to be much stronger in larger and longer lived animals and (ii) intrinsic physiological constraints on the complexity of tissue organization and potential lower limits on stem cell division rate. Neither the strength of selection nor the physiological constraints on tissue organization are known at present. However, in the case of the germ line mutation rate, which is proportional to the number of cell divisions in lineages leading to the gametes, current evidence indicates that physiological constraints are not limiting[44]. Across species, differences in effective population size, which is in general negatively correlated with body size and longevity[45], indicate the effectiveness of selection relative to drift. As a result, differences in effective population size between species determine the effectiveness of selection in spreading of favourable mutations and eliminating deleterious ones and, as such, can be used as indicator of the efficiency of selection[46,47]. This implies that, in contrast to somatic tissues, we expect germ line differentiation hierarchies to be more optimal for smaller animals with shorter life-spans as a result of their increased effective population sizes. For species for which information is available, the number of levels across species indeed follows an increasing trend as a function of the effective population size, ranging from $n = 5$ in humans with relatively small effective population size of $\sim 10^4$ and correspondingly less efficient selection, $n = 8$ in macaque with intermediate effective population size of the order of $10^5$, and $n = 10$ in mice with the largest effective population size of $\sim 5 \times 10^5$ (refs 48,49).

A qualitative examination of Fig. 3 suggests that a similar number of levels, of the order of $n \approx 10$ may be present in most somatic tissues, because the $D^*$ versus $n$ curve becomes progressively flatter after it reaches around twice the optimal value of $D^*$ at $n \gtrsim 10$, and the reduction in the divisional load becomes smaller and smaller as additional levels are added to the hierarchy and other factors are expected to limit further increase

in $n$. Alternatively, if we consider for example the human hematopoietic system, where $\sim 10^4$ hematopoietic stem cells (HSCs) produce a daily supply of $\sim 3.5 \times 10^{11}$ blood cells, we can calculate that over 80 years each stem cell produces a total of $N \approx 10^{12}$ terminally differentiated cells. For this larger value of $N$ the $D^*$ versus $n$ curve reaches twice the optimal value of $D^*$ at $n \gtrsim 15$ after which, similarly to Fig. 3, it becomes progressively flatter and the reduction in divisional load diminishes as additional levels are added. This rough estimate of $n \gtrsim 15$ levels is consistent with explicit mathematical models of human hematopoiesis that predict between 17 and 31 levels[14]. Active or short term HSCs (ST-HSCs) are estimated to differentiate about once a year, whereas a quiescent population of HSCs that provides cells to the active population is expected to be characterized by an even lower rate of differentiation. This is in good agreement with our prediction about the existence of a heterogeneous stem cell pool, a fraction of which consists of quiescent cells that only undergo a very limited number of cell cycles during the lifetime of the organism. Indeed, recently Busch et al.[6] found that adult hematopoiesis in mice is largely sustained by previously designated ST-HSCs that nearly fully self-renew, and receive rare but polyclonal HSC input. Mouse HSCs were found to differentiate into ST-HSCs only about three times per year.

For most somatic tissues the differentiation hierarchies that underpin the development of most cellular compartments remain inadequately resolved, the identity of stem and progenitor cells remains uncertain, and quantitative information on their proliferation rates is limited[20]. However, synthesis of available information on tissue organization by Tomasetti and Vogelstein[21], as detailed above, suggests that larger tissues with rapid turnover (for example, colon and blood) are relatively less prone to develop cancer. This phenomenon, as noted in the introduction, can be interpreted as Peto's paradox across tissues with the implication that larger tissues with rapid turnover rates have hierarchies with more levels and stem cells that divide at a slower pace. Accumulating evidence from lineage-tracing experiments[50] is also consistent with a relatively large number of hierarchical levels. Populations of stem cells in blood, skin, and the colon have begun to be resolved as combinations of cells that are long-lived yet constantly cycling, and emerging evidence indicates that both quiescent and active cell subpopulations may coexist in several tissues, in separate yet adjoining locations[41]. Lineage-tracing techniques[50] are rapidly developing, and may be used for directly testing the predictions of our mathematical model about the highly inhomogeneous distributions of the differentiation rates in the near future. In the context of estimates of the number of stem cells in different tissues that underlie Tomasetti and Vogelstein's results, the potential existence of such unresolved hierarchical levels suggests the possibility that the number of levels of the hierarchy are systematically underestimated and, correspondingly, that the number of stem cells at the base of these hierarchies are systematically overestimated.

Independent of the details of the hierarchy the dynamics of how divisional load accumulates in time is described by two phases: (i) a transient development phase during which each level of the hierarchy is filled up and (ii) a stationary phase during which homoeostasis is maintained in mature tissue. The dynamic details and the divisional load incurred during the initial development phase depend on the details of the hierarchy (cf. equations (5 and 8)). In contrast, in the stationary phase, further accumulation of the mutational load is determined solely by $\delta_0/N_0$ the rate at which tissue-specific stem cells differentiate at the bottommost level of the hierarchy. Such biphasic behaviour has been observed in the accumulation of mutations both in somatic[51] and germ line cells[1,52,53]. In both cases a substantial

number of mutations were found to occur relatively rapidly during development followed by a slower linear accumulation of mutation thereafter. General theoretical arguments imply that the contribution of the mutational load incurred during development to cancer risk is substantial[54], but this has been suggested to be in conflict with the fact that the majority of cancers develop late in life[51,55]. Resolving this question and more generally understanding the development of cancer in self-renewing tissues will require modelling the evolutionary dynamics of how hierarchical organization of healthy tissues breaks down.

Spontaneously occurring mutations accumulate in somatic cells throughout a person's lifetime, but the majority of these mutations do not have a noticeable effect. A small minority, however, can alter key cellular functions and a fraction of these confer a selective advantage to the cell, leading to preferential growth or survival of a clone[34]. Hierarchical tissue organization can limit somatic evolution at both these levels: (i) at the level of mutations, as we demonstrated above, it can dramatically reduce the number of cell divisions required and correspondingly the mutational load incurred during tissue homoeostasis; and (ii) at the level of selection acting on mutations with non-neutral phenotypic effects, as demonstrated by Nowak et al.[29] and later by Pepper et al.[31], tissues organized into serial differentiation experience lower rates of such detrimental cell-level phenotypic evolution. Extending the seminal results of Nowak et al. and Pepper et al., we propose that in addition to limiting somatic evolution at the phenotypic level, hierarchies are also how the tissues of multicellular organisms keep the accumulation of mutations in check, and that tissue-specific stem cells may in general correspond to a diverse set of slower dividing cell types.

In summary, we have considered a generic model of hierarchically organized self-renewing tissue, in the context of which we have derived universal properties of the divisional load during tissue homoeostasis. In particular, our results provide a lower bound for the lifetime divisional load of a tissue as a function of the number of its hierarchical levels. Our simple analytical description provides a quantitative understanding of how hierarchical tissue organization can limit unwanted somatic evolution, including cancer development. Surprisingly, we find that the theoretical minimum number of cell divisions can be closely approached (cf. Fig. 3, where the theoretical minimum corresponds to the dashed horizontal line), demonstrating that hierarchical tissue organization provides a robust and nearly ideal mechanism to limit the divisional load of tissues and, as a result, minimize somatic evolution.

**Data availability**. No data was generated as part of this study.

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

## Acknowledgements

This work was supported by the Hungarian Science Foundation (grant K101436). We would like to acknowledge the comments of anonymous reviewers on a previous version of the manuscript, as well as discussion with and comments from Bastien Boussau, Márton Demeter, Máté Kiss and Dániel Grajzel.

## Author contributions

I.D. and G.J.S. designed the study, carried out research and wrote the paper.

## Additional information

**Competing financial interests:** The authors declare no competing financial interests.

**Publisher's note**: 

