## [Peer review file · Nature Communications]

Reviewers' comments:

Reviewer #1 (Remarks to the Author):

The authors present a clear and meaningful mathematical analysis that describes key aspects of mammalian tissue homeostasis and, potentially, tumor evolution. Based on a set of thoughtful assumptions, the authors derive simple and intuitive relations between key determinants of a hierarchically structured, self-renewing cell population in equilibrium and a newly defined quantity called divisional load of the tissue/organism. This provides quantitative grounds for the deep connection between such organization principles and disease prevention. These quantitative grounds are necessary in the emerging field of systems medicine. I can generally see that this manuscript, which is mostly clearly written and in good style, can be of interest to a wider audience, and thus be fit for Nature Communications. At the same time, these findings can provide quantitative foundations for novel experimental or even clinical approaches to better understand, e.g., hematopoiesis and stem cell driven dynamics. However, I urge the authors to address a few important points which I attached as embedded comments to the pdf, as well as one general thought: can the authors provide a meaningful connection to empirical evidence, either existing or from experiments to be made by others? Only when such beautiful mathematical results can directly relate to experimental or clinical findings in form of reproducible data (eg biomarkers in disease progression, molecular profiles in a patient population, or genetic/epigenetic insights from a mouse model). Recent manuscripts that come to mind that combine math modeling and empirical data the papers by Werner and colleagues (eLife 2015; Cancer Research 2016), or Tang, Zhao et al, Clinical Cancer Research 2016 (PMID: 27006493). I suggest major revisions along these lines.

Reviewer #2 (Remarks to the Author):

Review of: Hierarchical tissue organization as a general mechanism to limit somatic evolution, by Imre Derényi and Gergely Szöllósi.

In their article under consideration for Nature Communications, Derényi and Szöllósi discuss properties of a hierarchical tissue organization that minimise the number of cell divisions per cell lineage for the life time of an organism. They find that the number of cell divisions per lineage increases linearly in time driven by stem cell divisions and the optimal architecture is independent of the detailed structure of the hierarchy. These results are interesting and important to further increase our understanding of cancer initiation and progression. There are a few points, the authors should address, before the manuscript might be accepted for publication in Nature Communication.

The most critical point in my point of view is summarized in equation (15). This equation states, that the optimal tissue architecture has a depth n of $n = \log_2 \left[\frac{N}{10} \right] + 0.471$, where N represents the total number of fully differentiated cells produced in the lifetime of an organism. This seems to lead to unrealistic estimates for n especially in tissues with high turnover. For example, humans produce approximately 2×10^{11} red blood cells a day, which would amount to $\approx 10^{16}$ produced red blood cells after 80 years and a corresponding hierarchy depth of $n \approx 53$. This seems very high.

In addition, the predicted number of stem cell divisions in an organism in homeostasis would be less than 2 in the total lifetime of the organism. Again, this seems very extreme. Estimates of stem cell turn over in actual human tissue varies with tissue architecture and is in the order of once every 4 days in colon and maybe 1 or 2 divisions per year in hematopoietic stem cells. Which additional mechanisms rather than mutation accumulation would prevent the evolution of this optimal architecture? The authors should improve their discussion on this aspect and fairly compare their predictions to known quantities in actual tissues.

It is likely that the tissue architecture is determined by multiple factors, for example, the robustness of the stem cell pool against mutation invasion or stochastic extinction, the flexibility of the hierarchy to repair tissue damage or the robustness to stochastic oscillations. At least, this should be discussed.

In order to calculate the optimal tissue architecture, the authors neglected cell death. However, cell death seems to be unavoidable and occurs at any stage of the hierarchy. How robust are the results, if the authors would consider cell death at different stages of the hierarchy?

Reviewer #3 (Remarks to the Author):

The role of tissue organization and cell differentiation in limiting somatic evolution and cancer is an important topic. The analytical math model present here is new, and some of its results are probably useful and worth publishing. I recommend a major revision.

The central problem with this manuscript is that it presents as novel, several ideas and results that have been published before. Both the Title and Conclusions section are problematic in this regard. The title is "Hierarchical tissue organization as a general mechanism to limit somatic evolution". The final sentence of the Conclusions states that, "We propose that hierarchies are how the tissues of multicellular organisms keep the accumulation of mutations in check... and that tissue-specific stem cells may in general correspond to a diverse set of slower dividing cell types." All of this is correct, but fails to acknowledge that these ideas are already well-established. The quote from the Conclusion fails to recognize that this hypothesis was not only proposed previously by Pepper et al. (2007), but was also tested and validated by those authors using an experimental *in silico* evolutionary system. Most egregiously, the abstract states that, "Although qualitative arguments suggesting that hierarchically organized tissues may be optimal in reducing somatic evolution go back several decades, a quantitative explanation has been lacking". This is absolutely not true, although this type of analytical math model has been lacking. Similarly, the present submission is incorrect in stating that, "The potential of hierarchical tissues to limit somatic mutational load simply by reducing the number of cell divisions along cell lineages... has not been explored". It was thoroughly explored in the above-cited publication. Other previous work is partially recognized in the paragraph beginning with, "Models conceptually similar to ours have a long history...", but that does not solve the problem that the central ideas and results presented here are not new, though that claim is often stated or implied.

To the extent that the current manuscript makes new contributions, they concern not new ideas, but the analytical math model presented, and certain of its formal results. These narrower results should be highlighted and emphasized, and any explicit or implied claims to be the first with the broader ideas or hypotheses should be removed. Buried in the body text is a suggested focus on, "the uneven distribution of divisional rates across the differentiation hierarchy." This focus is lost in the rest of the manuscript, and in the over-broad implications of new ideas.

Given that some of the questions addressed in the current manuscript were previously addressed by Pepper et al. (2007), using an agent-based computational model, it should be clarified not only what results are new here, but also what advantage is provided by the current analytical modelling method over what was done previously with an agent-based model.

Organization and presentation

Most of the section under the heading 'RESULTS' would be better described as 'The Model', as it actually consists descriptions of the model assumptions and the notation used. However, some elements of this section follow from the assumptions in non-trivial ways, and might be useful to highlight. These could be usefully extracted and collected in a 'RESULTS' section. These include equations (9), (10), and (13) through (17), as well as the first two paragraphs of the 'DISCUSSION' section.

Cited ref:

JW Pepper, K Sprouffske, and CC Maley. (2007). Animal Cell Differentiation Patterns Suppress Somatic Evolution PloS Computational Biology 3(12):2532-2545.

Reply to reviewers' comments

Reviewer #1 (Remarks to the Author):

The authors present a clear and meaningful mathematical analysis that describes key aspects of mammalian tissue homeostasis and, potentially, tumor evolution. Based on a set of thoughtful assumptions, the authors derive simple and intuitive relations between key determinants of a hierarchically structured, self-renewing cell population in equilibrium and a newly defined quantity called divisional load of the tissue/organism. This provides quantitative grounds for the deep connection between such organization principles and disease prevention. These quantitative grounds are necessary in the emerging field of systems medicine. I can generally see that this manuscript, which is mostly clearly written and in good style, can be of interest to a wider audience, and thus be fit for Nature Communications. At the same time, these findings can provide quantitative foundations for novel experimental or even clinical approaches to better understand, e.g., hematopoiesis and stem cell driven dynamics. However, I urge the authors to address a few important points which I attached as embedded comments to the pdf, as well as one general thought: can the authors provide a meaningful connection to empirical evidence, either existing or from experiments to be made by others? Only when such beautiful mathematical results can directly relate to experimental or clinical findings in form of reproducible data (eg biomarkers in disease progression, molecular profiles in a patient population, or genetic/epigenetic insights from a mouse model). Recent manuscripts that come to mind that combine math modeling and empirical data the papers by Werner and colleagues (eLife 2015; Cancer Research 2016), or Tang, Zhao et al, Clinical Cancer Research 2016 (PMID: 27006493). I suggest major revisions along these lines.

Reply: In Page 8 we now include a long discussion about the connection of our results to the empirical properties of the hematopoietic system. We have also noted that as lineage-tracing techniques are rapidly developing, they may be used for directly testing the predictions of our mathematical model in the near future.

Reviewer #1's points attached as embedded comments in the pdf of our manuscript:

Page 2: In addition, the mere existence of the hierarchical structure can be exploited to reveal properties of tissue aging, disease and cancer treatment, see eg Werner et al eLife 2016, and Werner et al Cancer Research 2016, and recent work by Komarova's group in PLoS Comp Biol.

Reply: We agree completely with Reviewer #1's point and discuss the implications of our results both in the discussion, which we have extended in the revised version of the manuscript, and more briefly at the end of the introduction where we now write, citing the above mentioned papers:

“More generally, evolutionary thinking is becoming an indispensable tool to understand cancer, and even to propose directions in the search for treatment strategies[36]. Models that integrate

information on tissue organization have not only provided novel insight into cancer as an evolutionary process [27, 37], but have also produced direct predictions for improved treatment[38–40]. The simple and intuitive relations that we derive below have the potential to further this field of research by providing quantitative grounds for the deep connection between organization principles of tissues and disease prevention and treatment.”

Page 2: Re:Tomasetti & Vogelstein. Mention here that especially the assumed numbers of stem cells in each tissue might be a significant source of error. Could your work contribute to systematic approaches that can elucidate these numbers more specifically?

Reply: One of the primary conclusions of our results is that increasing the number of hierarchical levels can greatly reduce the number of cell divisions along cell lineages (i.e. divisional load). Furthermore, we find that a relatively large number of hierarchical levels are needed to achieve the minimum possible divisional load (see equation 15), with the relative benefit of adding additional levels progressively declining as the optimum is approached (see Fig. 3 and the fourth and fifth paragraph of the discussion on pages 7 and 8). In the context of Tomasetti and Vogelstein’s results our findings suggest, as we now write in page 8, that the number of levels of the hierarchy are potentially underestimated and correspondingly that the number of stem cells at the base of these hierarchies are overestimated:

“For most somatic tissues the differentiation hierarchies that underpin the development of most cellular compartments remain inadequately resolved, the identity of stem and progenitor cells remains uncertain, and quantitative information on their proliferation rates is limited [20]. However, synthesis of available information on tissue organization by Tomasetti and Vogelstein [21], as detailed above, suggests that larger tissues with rapid turnover (e.g., colon and blood) are relatively less prone to develop cancer. This phenomenon, can be interpreted as Peto’s paradox across tissues with the implication that larger tissues with rapid turnover rates have hierarchies with more levels and stem cells that divide at a slower pace. Accumulating evidence from cell lineage-tracing experiments in these tissues [49] is also consistent with a relatively large number of hierarchical levels. Populations of stem cells in blood, skin, and the colon have begun to be resolved as combinations of cells that are long-lived yet constantly cycling, and emerging evidence indicates that both quiescent and active cell subpopulations may coexist in several tissues, in separate yet adjoining locations [28]. **Lineage-tracing techniques [49] are rapidly developing, and may be used for directly testing the predictions of our mathematical model about the highly inhomogeneous distributions of the differentiation rates in the near future.**

In the context of estimates of the number of stem cells in different tissues that underlie the Tomasetti and Vogelstein’s results, the potential existence of such unresolved hierarchical levels suggests the possibility that number of levels of the hierarchy are systematically underestimated and correspondingly that the number of stem cells at the base of these hierarchies are systematically overestimated.”

Page 3: In what way can such modeling help to improve therapy, see recent reviews Altrock et al, Nature Revs Cancer, 2015; Michor & Beal Cell 2015

Reply: We believe that our results have the potential to serve as a theoretical foundation for future research by providing a general quantitative model of homeostasis and, by extension, somatic evolution in hierarchically organized tissue. Our proposal that hierarchical tissue organization has evolved as a mechanism to minimize divisional load underscores the importance of modeling tissue organization if we wish to understand how tumours evolve from healthy tissues. In particular, instead of simply modeling “driver” mutations that increase net cell division rate we have to model mutations that change the rate of the elementary processes that implement the differentiation hierarchy. Only in this way, by understanding which type of mutations lead to the breakdown of tissue homeostasis, can we gain insight into the possible pathways that lead to cancer.

It has already been demonstrated by Franziska Michor pioneering work that taking into consideration the simplest possible hierarchical organization in modelling tumour response can impact cancer treatment. A more nuanced and model based understanding of the hierarchical organization of tumors can be expected to provide further treatments improvements and insight.

Page 3: What do you mean by "ideal"? Is there some form of optimization process in play?

Reply: By “*nearly ideal mechanism to limit the divisional load of tissues*” we mean a mechanism that approaches closely the theoretical minimum number of cell divisions required (cf. first paragraph of the introduction).

Page 3: Rewrite for clarity

Reply: Corrected, we now write:

“We argue that hierarchies are how the tissues of multicellular organisms keep the accumulation of mutations in check, and that populations of cells currently believed to correspond to tissue-specific stem cells may in general constitute a diverse set of slower dividing cell types [6, 28].”

Page 3: Shouldn't the cell compartments divide progressively faster as cells become more differentiated? Please clarify.

Reply: Yes that is correct. We have rephrased the sentence in the revised version of the manuscript to clarify, we now write:

“As long as a sufficient number of progressively slower dividing cell types towards the root of the hierarchy are present, optimal self-sustaining differentiation hierarchies can produce N terminally differentiated cells during the course of an organism’s lifetime from a single precursor with no more than $\log_2(N) + 2$ cell divisions along any lineage.”

Page 3: But is the power law the only meaningful analytical structure here? Can we justify it via the underlying stochastic processes?

Reply: Werner et al. (refs. 33, 34) assume a power law *a priori*, but we do not make any such assumption. Our analytical results, however, show (see equation 11) that differentiation rates in optimal hierarchies that minimize divisional load do in fact follow a power law. In this sense the use of a power law by Werner et al. is *a posteriori* justified by our analysis of the underlying stochastic process.

Page 4: Maybe include a few pages supplementary materials where you derive this approximation from first principles.

Reply: We agree with that the derivation of equation 2 could have been better described. After careful consideration, and in light of journal policy requesting that we avoid supplementary material if possible, we decided that we could best include such clarification as part of figure 2 and the accompanying legend, as well as the main text in page 4:

FIG. 2. **Hierarchical cell differentiation in self-renewing tissue.** a) a model tissue produces terminally differentiated cells through n intermediate levels of partially differentiated cells. b) five microscopic events can occur with a cell (i) symmetric cell division with differentiation, (ii) asymmetric cell division, (iii) symmetric cell division without differentiation, (iv) single cell differentiation, and (v) cell death. To the right of each event type we give the corresponding per cell rate of cell division used to derive equation 2.

..

the stem cell level was created at time zero. Using the rates of the microscopic events (also shown in Fig. 2b), considering that each division increases the accumulated number of divisions of both daughter cells by one, and taking into account the divisional loads that the departure of cells take and the arrival of cells bring, the following mean-field differential equation system can be formulated for the time evolution of the total divisional load ($D_k N_k$) of levels $k < n$ of a fully developed tissue:

$$\begin{aligned} \dot{D}_k N_k = & -\frac{\delta_k}{2} p_k D_k + \delta_k (1-p_k) \\ & + \frac{\delta_k}{2} p_k [q_k (D_{k-1} + 1) + (1-q_k)(D_k + 2)] . \end{aligned} \quad (2)$$

Page 4: Ambiguous “this”

Reply: Corrected.

Page 7: Citation and rationale?

Reply: We added further citations and expanded on the rationale, writing on page 8 of the revised version of the manuscript:

“Across species differences in effective population size, which is in general negatively correlated with body size and longevity [44], indicate the effectiveness of selection relative to drift. As a result, differences in effective population size between species determine the effectiveness of selection in spreading of a favourable mutations and eliminating deleterious ones, and as such can be used as indicator of the efficiency of selection [45,46].”

Page 8: This seems very high (in %)

Reply: We have removed this section, as our model does not directly offer a prediction of the fraction of stem cells.

Page 8: Unclear what closely approached may mean

Reply: In the revised version of the manuscript we added a reference to figure 3 to clarify:

“Surprisingly, we find that the theoretical minimum number of cell divisions can be very closely approached (cf. Fig.3, where the theoretical minimum corresponds to the dashed horizontal line), demonstrating that hierarchical tissue organization provides a robust and nearly ideal mechanism to limit the divisional load of tissues and, as a result, minimize somatic evolution.”

Page 8: Elaborate in an additional sentence what minimize means here

Reply: We elaborate in the following paragraph where we write:

“Spontaneously occurring mutations accumulate in somatic cells throughout a person’s lifetime, but the majority of these mutations do not have a noticeable effect. A small minority, however, can alter key cellular functions and a fraction of these confer a selective advantage to the cell, leading to preferential growth or survival of a clone [35]. Hierarchical tissue organization can limit somatic evolution at both these levels: (i) at the level of mutations, as we demonstrated above, it can dramatically reduce the number of cell divisions required and correspondingly the mutational load incurred during tissue homeostasis; and (ii) ..”

Page 8: Can you comment on how your model findings can be tested empirically? What is the immediate experimental link, how would you come up with a testable hypothesis?

Reply: Testing our model predictions would require detailed information on the life history of populations of cells coming from healthy tissues and potentially also tumors. In order to recover the most information about the life history of a population of cells we would need to reconstruct the bifurcating lineage tree the root of which is the zygote. The reconstruction of such phylogenies, however, would require high-quality single cell sequencing data from a large number of cells. Fortunately, the technologies for single cell sequencing have been advancing at a rapid pace in the last few years (see e.g. the 2016 review in Genome Research by Nicolas Navin).

As more and better quality single cell sequencing data become available we envision reconstructing phylogenetic trees with methods that use an explicit model of the underlying evolutionary process based on the model we presented in our manuscript under review. Such an approach will allow us to infer directly from genomic sequence data parameters, such as the number of levels of the tissue hierarchy, that at present we can only predict based on optimization arguments.

We also note in the manuscript that:

“Lineage-tracing techniques [49] are rapidly developing, and may be used for directly testing the predictions of our mathematical model about the highly inhomogeneous distributions of the differentiation rates in the near future.”

Reviewer #2 (Remarks to the Author):

In their article under consideration for Nature Communications, Derényi and Szöllósi discuss properties of a hierarchical tissue organization that minimise the number of cell divisions per cell lineage for the life time of an organism. They find that the number of cell divisions per lineage increases linearly in time driven by stem cell divisions and the optimal architecture is independent of the detailed structure of the hierarchy. These results are interesting and important to further increase our understanding of cancer initiation and progression. There are a few points, the authors should address, before the manuscript might be accepted for publication in Nature Communication.

The most critical point in my point of view is summarized in equation (15). This equation states, that the optimal tissue architecture has a depth n of $n = \log_2 N + 0.471$, where N represents the total number of fully differentiated cells produced in the lifetime of an organism. This seems to lead to unrealistic estimates for n especially in tissues with high turnover. For example, humans produce approximately 2×10^{11} red blood cells a day, which would amount to $\approx 10^{16}$ produced red blood cells after 80 years and a corresponding hierarchy depth of $n \approx 53$. This seems very high.

Reply: The Reviewer highlights an important points that were not sufficiently clear in the previous version of our manuscript: First of all, the results we present, including equation (15) highlighted in the review, are given per stem cell. In the case of equation (15) this means that the optimal number of hierarchical levels n_{opt} is given for a total of N terminally differentiated cells produced by a single initial stem cell. Considering that there are an estimated $\approx 10\,000$ haemopoietic stem cells we have $N \approx 10^{12}$ red blood cells per stem cell. According to equation (15) this correspond to $n_{opt} \approx 40$, a number that is still arguably too high to be physiologically realistic.

This brings us to the second point: how closely can we expect selection to drive real tissues to optimal ones? Setting aside physiological constraints on tissue architecture, evolutionary theory provides some broad answers: it is clear that selection is practically powerless to combat causes of death after reproductive age, thus mechanisms are expected to primarily evolve to reduce cancer in individuals before the end of reproductive age. Furthermore, the efficacy of even these mechanisms will be limited by genetic drift, which makes cancer incidence in pre-reproductive age individuals unlikely to be reduced below the reciprocal of the effective population size, i.e. $\sim 1/10,000$ in humans. Finally, taking into consideration the dependence of divisional load on the number of hierarchical levels plotted below for $N \approx 10^{12}$ we can see that, similar to Figure 3 in the manuscript, where we assume a different value of N , the reduction in divisional load as n increases levels off after $n \approx 15-20$. This suggests that limits to the efficiency of selection have kept the number of levels in the haemopoietic system at around $n = 15-20$, a number that can be reconciled with our current understanding of the physiology of human

hematopoiesis, explicit modeling of which suggests an architectural organization comprised of between 17 to 31 stages (cf. Dingli et al. 2007 PLoS One and refs. [6,12,13] therein).

To clarify this point in the revised version of the manuscript we now write:

“A qualitative examination of Fig. 3 suggests that a similar number of levels, of the order of $n \approx 10$ may be present in most somatic tissues, because the D^* vs. n curve becomes progressively flatter after it reaches around twice the optimal value of D^* at $n \geq 10$, and the reduction in the divisional load becomes smaller and smaller as additional levels are added to the hierarchy and other factors are expected to limit further increase in n . Alternatively, if we consider for example the human hematopoietic system, where approximately 10^4 hematopoietic stem cells (HSCs) produce a daily supply of $\sim 3.5 \times 10^{11}$ blood cells, we can calculate that over 80 years each stem cell produces a total of $N \approx 10^{12}$ terminally differentiated cells. For this larger value of N the D^* vs. n curve reaches twice the optimal value of D^* at $n \geq 15$ after which, similarly to Fig. 3, it becomes progressively flatter and the reduction in divisional load diminishes as additional levels are added. This rough estimate of $n \geq 15$ levels is consistent with explicit mathematical models of human hematopoiesis that predict between 17 and 31 levels [14].“

In addition, the predicted number of stem cell divisions in an organism in homeostasis would be less than 2 in the total lifetime of the organism. Again, this seems very extreme. Estimates of stem cell turn over in actual human tissue varies with tissue architecture and is in the order of once every 4 days in colon and maybe 1 or 2 divisions per year in hematopoietic stem cells. Which additional mechanisms rather than mutation accumulation would prevent the evolution of this optimal architecture? The authors should improve their discussion on this aspect and fairly compare their predictions to known quantities in actual tissues.

Reply: The Reviewer correctly points out that our model does not consider the potentially varied physiological constraints affecting different tissues. However, as we elaborated above, we did not mean to suggest that real-life tissues possess an ideal hierarchical organization that fully minimizes divisional load. Rather, our results we believe demonstrate that hierarchical demonstration offers a general mechanism to limit the accumulation of somatic mutations by minimizing divisional load. In the context of the above example of haematopoiesis one of the reasons for there being up to 30 levels, instead of the 40 that we estimate above to be optimal, is that the real-life hematopoietic system is not fully optimal, e.g. because hematopoietic stem cells divide more than twice during an individual's lifetime.

It is likely that the tissue architecture is determined by multiple factors, for example, the robustness of the stem cell pool against mutation invasion or stochastic extinction, the flexibility of the hierarchy to repair tissue damage or the robustness to stochastic oscillations. At least, this should be discussed.

Reply: We agree with the Reviewer that in addition to the accumulation of mutations, both the robustness of stem cell pool against stochastic extinction and population oscillations, and the problem of mutation invasion, which concerns somatic evolution at the phenotypic level are important issues, which we now mention at the end of the results section:

“This means, that the minimization of the divisional load poses no constraint on the number of stem cells. However, since both maintaining a larger number of differentiation levels and keeping the differentiation hierarchy closer to optimum involve more complicated regulation, we suspect that a relatively large stem cell pool is beneficial, especially as a larger stem cell population can also be expected to be more robust against stochastic extinction, population oscillation, and injury.”

and the conclusions:

“Spontaneously occurring mutations accumulate in somatic cells throughout a person’s lifetime, but the majority of these mutations do not have a noticeable effect. A small minority, however, can alter key cellular functions and a fraction of these confer a selective advantage to the cell, leading to preferential growth or survival of a clone [35]. Hierarchical tissue organization can limit somatic evolution at both these levels: (i) at the level of mutations, as we demonstrated above, it can dramatically reduce the number of cell divisions required and correspondingly the

mutational load incurred during tissue homeostasis; and (ii) at the level of selection acting on mutations with non-neutral phenotypic effects, as demonstrated by Nowak et al. [30] and later by Pepper et al. [32], tissues organized into serial differentiation experience lower rates of such detrimental cell-level phenotypic evolution. Extending the seminal results of Nowak et al. and Pepper et al., we propose that in addition to limiting somatic evolution at the phenotypic level, hierarchies are also how the tissues of multicellular organisms keep the accumulation of mutations in check, and that tissue-specific stem cells may in general correspond to a diverse set of slower dividing cell types.“

In order to calculate the optimal tissue architecture, the authors neglected cell death. However, cell death seems to be unavoidable and occurs at any stage of the hierarchy. How robust are the results, if the authors would consider cell death at different stages of the hierarchy?

Reply: The Reviewer raises a very important point. The reason that we did not consider cell death is as we write that

“Our goal is to determine the optimal tissue organization and dynamics that minimize the number of cell divisions that the cells undergo until they become terminally differentiated. For this reason cell death, except for the continuous expenditure of terminally differentiated cells, is disallowed as it can only increase the number of divisions.”

The Review is correct, however, in that we did not discuss how introducing cell would affect divisional load. We now write in the revised version of the manuscript:

“We note, however, that cell death with a rate proportional to that of cell divisions would simply result in a proportionally increased divisional load and, thus, would have no effect on the optimum.”

Reviewer #3 (Remarks to the Author):

The role of tissue organization and cell differentiation in limiting somatic evolution and cancer is an important topic. The analytical math model present here is new, and some of its results are probably useful and worth publishing. I recommend a major revision.

The central problem with this manuscript is that it presents as novel, several ideas and results that have been published before. Both the Title and Conclusions section are problematic in this regard. The title is “Hierarchical tissue organization as a general mechanism to limit somatic evolution”. The final sentence of the Conclusions states that, “We propose that hierarchies are how the tissues of multicellular organisms keep the accumulation of mutations in check... and that tissue-specific stem cells may in general correspond to a diverse set of slower dividing cell types.” All of this is correct, but fails to acknowledge that these ideas are already well-established. The quote from the Conclusion fails to recognize that this hypothesis was not only proposed previously by Pepper et al. (2007), but was also tested and validated by those authors using an experimental in silico evolutionary system. Most egregiously, the abstract states that, “Although qualitative arguments suggesting that hierarchically organized tissues may be optimal in reducing somatic evolution go back several decades, a quantitative explanation has been lacking”. This is absolutely not true, although this type of analytical math model has been lacking. Similarly, the present submission is incorrect in stating that, “The potential of hierarchical tissues to limit somatic mutational load simply by reducing the number of cell divisions along cell lineages... has not been explored”. It was thoroughly explored in the above-cited publication. Other previous work is partially recognized in the paragraph beginning with, “Models conceptually similar to ours have a long history...”, but that does not solve the problem that the central ideas and results presented here are not new, though that claim is often stated or implied.

To the extent that the current manuscript makes new contributions, they concern not new ideas, but the analytical math model presented, and certain of its formal results. These narrower results should be highlighted and emphasized, and any explicit or implied claims to be the first with the broader ideas or hypotheses should be removed. Buried in the body text is a suggested focus on, “the uneven distribution of divisional rates across the differentiation hierarchy.” This focus is lost in the rest of the manuscript, and in the over-broad implications of new ideas. Given that some of the questions addressed in the current manuscript were previously addressed by Pepper et al. (2007), using an agent-based computational model, it should be clarified not only what results are new here, but also what advantage is provided by the current analytical modelling method over what was done previously with an agent-based model.

Reply: We agree with the Reviewer that we have neglected to discuss the seminal contribution by Pepper et al. published in PLoS Computational Biology in 2007. We remedy this shortcoming in the revised version of the manuscript, as described below.

Pepper et al. used an agent-based computer simulation of cell population dynamics and evolution within tissues to demonstrate that, relative to other, simpler tissue organization patterns, hierarchically organized tissues experience lower rates of detrimental cell-level phenotypic evolution. Our results are different in a very important respect. As we elaborate in the conclusion section of the revised version of the manuscript, we consider somatic evolution at the level of mutation accumulation, while Pepper et al.'s results (citation [32]) concern the level of selection acting on mutations with non-neutral phenotypic effects:

“Spontaneously occurring mutations accumulate in somatic cells throughout a person’s lifetime, but the majority of these mutations do not have a noticeable effect. A small minority, however, can alter key cellular functions and a fraction of these confer a selective advantage to the cell, leading to preferential growth or survival of a clone [35]. Hierarchical tissue organization can limit somatic evolution at both these levels: (i) at the level of mutations, as we demonstrated above, it can dramatically reduce the number of cell divisions required and correspondingly the mutational load incurred during tissue homeostasis; and (ii) at the level of selection acting on mutations with non-neutral phenotypic effects, as demonstrated by Nowak et al. [30] and later by Pepper et al. [32], tissues organized into serial differentiation experience lower rates of such detrimental cell-level phenotypic evolution. Extending the seminal results of Nowak et al. and Pepper et al., we propose that in addition to limiting somatic evolution at the phenotypic level, hierarchies are also how the tissues of multicellular organisms keep the accumulation of mutations in check, and that tissue-specific stem cells may in general correspond to a diverse set of slower dividing cell types.”

To clarify this point we propose the possibility of changing the title of the manuscript from “*Hierarchical tissue organization as a general mechanism to limit somatic evolution*” to “*Hierarchical tissue organization as a general mechanism to limit the accumulation of somatic mutations*”, but would prefer to leave the final decision in this matter to the editor.

We have also made corresponding changes in the first sentence of the abstract:

“How can tissues generate large numbers of cells, yet keep the divisional load (the number of divisions along cell lineages) low in order to curtail the accumulation of somatic mutations and reduce the risk of cancer?”

As well as corresponding changes in the introduction:

“We show that in self-renewing tissues hierarchical organization provides a robust and nearly ideal mechanism to limit the divisional load of tissues and, as a result, minimize the accumulation of somatic mutations that fuel somatic evolution and can lead to cancer.”

We also extended our discussion of the literature to include Pepper et al.'s work (citation [32]):

“More recently a series of studies have investigated the dynamics of mutations in hierarchical tissues with dynamical asymmetry [32–34] and found that hierarchical tissue organization can (i) suppress single [33] as well as multiple mutations [34] that arise in progenitor cells, and (ii) slow down the rate of somatic evolution towards cancer [32] if selection on mutations with non-neutral phenotypic effects is also taken into account.”

Organization and presentation

Most of the section under the heading ‘RESULTS’ would be better described as ‘The Model’, as it actually consists descriptions of the model assumptions and the notation used. However, some elements of this section follow from the assumptions in non-trivial ways, and might be useful to highlight. These could be usefully extracted and collected in a ‘RESULTS’ section. These include equations (9), (10), and (13) through (17), as well as the first two paragraphs of the ‘DISCUSSION’ section.

Reply: We appreciate the Reviewer’s suggestions, but after reviewing the formatting guidelines (cf. <http://www.nature.com/ncomms/submit/content-types>), we believe that our formatting better fits the instructions therein.

REVIEWERS' COMMENTS:

Reviewer #1 (Remarks to the Author):

The authors have nicely answered all my concerns.

one last remark: Citation 37 (Waclav et al, Nature) has not much to do with what the authors are trying to say in this paragraph or the paper in general, as [37] is about mutation accumulation in spatial growth. Concerning tissue structure/organization and modeling, cite Rejniak & Anderson, "Hybrid models of tumor growth" (2010) and Altrock et al., "[OBJ]The mathematics of cancer: integrating quantitative models" (2015) as all relevant information is in these.

Reviewer #2 (Remarks to the Author):

In believe this manuscript is a nice contribution to the theoretical understanding of tissue organisation in general and hierarchical tissues in particular. As it is a more conceptual manuscript, the authors cannot necessarily provide a practical advise to treat cancer, but I don't see this as a short coming.

The authors addressed all my suggestions and I think the manuscript can be readily published in Nature Communications.

I suggest publication at this stage.

Reviewer #3 (Remarks to the Author):

Most of my concerns have been adequately addressed. I will not insist that the remaining problems should prevent acceptance, but I do flag this as a question for the editor to consider. The two main remaining problems are:

1) The authors still take credit for creating the central ideas presented here, which are not theirs. Only the specific mathematical treatment used here is novel. Clearly, to these math-oriented authors, an idea has not been presented until the math has been written out, but most scientists consider verbal logic and computational models to be valid as well.

2) The organization remains awkward. In particular, the 'RESULTS' section consists not of results, but mostly of descriptions of the of model assumptions math notation, while the

actual results of the analysis are scattered in this section and in the 'Discussion' section. I will not insist that this problem should prevent acceptance, but I do flag it as a question for the editor to consider.

Reply to reviewers' comments

Reviewer #1 (Remarks to the Author):

The authors have nicely answered all my concerns.

One last remark: Citation 37 (Waclav et al, Nature) has not much to do with what the authors are trying to say in this paragraph or the paper in general, as [37] is about mutation accumulation in spatial growth. Concerning tissue structure/organization and modeling, cite Rejniak & Anderson, "Hybrid models of tumor growth" (2010) and Altrock et al., "The mathematics of cancer: integrating quantitative models" (2015) as all relevant information is in these.

Reply: We have made the requested changes.

Reviewer #2 (Remarks to the Author):

In believe this manuscript is a nice contribution to the theoretical understanding of tissue organisation in general and hierarchical tissues in particular. As it is a more conceptual manuscript, the authors cannot necessarily provide a practical advise to treat cancer, but I don't see this as a short coming.

The authors addressed all my suggestions and I think the manuscript can be readily published in Nature Communications.

I suggest publication at this stage.

Reviewer #3 (Remarks to the Author):

Most of my concerns have been adequately addressed. I will not insist that the remaining problems should prevent acceptance, but I do flag this as a question for the editor to consider. The two main remaining problems are:

1) The authors still take credit for creating the central ideas presented here, which are not theirs. Only the specific mathematical treatment used here is novel. Clearly, to these math-oriented authors, an idea has not been presented until the math has been written out, but most scientists consider verbal logic and computational models to be valid as well.

Reply: We describe the extent to which are results our novel in the introduction:

“Qualitative arguments suggesting that hierarchically organized tissues may be optimal in reducing the accumulation of somatic mutations go back several decades [27]. As mutations provide the fuel for somatic evolution (including not only the development of cancer, but also tissue degeneration, aging, germ line deterioration, etc.) it is becoming widely accepted that tissues have evolved to minimize the accumulation of somatic mutations during the lifetime of an individual [27]. The potential of hierarchical tissues to limit somatic mutational load simply by reducing the number of cell divisions along cell lineages, however, has not been explored in a mathematically rigorous way.”

To further clarify this point we have revised the manuscript in several places:
In the abstract we now write:

“.. we consider a general model of hierarchically organized self-renewing tissues and show that the lifetime divisional load of such a tissue is independent of the details of the cell differentiation processes, and depends only on two structural and two dynamical parameters.”, instead of “.. we introduce a general model of hierarchically organized self-renewing tissues .. ”.

Similarly, in the results section we now write:

“To quantify how many times the cells of self-renewing tissues undergo cell divisions during tissue development and maintenance, we consider a minimal generic model of hierarchically organized, self-sustaining tissue.”

Finally, in the discussion we now write

“In summary, we have considered a generic model of hierarchically organized self-renewing tissue, in the context of which we have derived universal properties of the divisional load during tissue homeostasis. In particular, our results provide a lower bound for the lifetime divisional load of a tissue as a function of the number of its hierarchical levels.”, instead of “.. we have introduced a generic model of hierarchically organized self-renewing tissue .. ”.

2) The organization remains awkward. In particular, the ‘RESULTS’ section consists not of results, but mostly of descriptions of the of model assumptions math notation, while the actual results of the analysis are scattered in this section and in the ‘Discussion’ section. I will not insist that this problem should prevent acceptance, but I do flag it as a question for the editor to consider.

Reply: After rereading our manuscript we agree with the criticism of the organization by reviewer #3. To remedy these problems we have moved parts of the the discussion to the results section and structured the results section into three subsections.